# Synthesis and Cytotoxicity Study of Magnetite Nanoparticles Coated with Polyethylene Glycol and Sorafenib–Zinc/Aluminium Layered Double Hydroxide

**DOI:** 10.3390/polym12112716

**Published:** 2020-11-17

**Authors:** Mona Ebadi, Kalaivani Buskaran, Saifullah Bullo, Mohd Zobir Hussein, Sharida Fakurazi, Giorgia Pastorin

**Affiliations:** 1Materials Synthesis and Characterization Laboratory, Institute of Advanced Technology (ITMA), Universiti Putra Malaysia, Serdang 43400, Malaysia; mona.ebadi64@gmail.com (M.E.); bullosaif1@gmail.com (S.B.); 2Laboratory of Vaccine and Immunotherapeutics, Institute of Bioscience, Universiti Putra Malaysia, Serdang 43400, Malaysia; vaneey88@yahoo.com (K.B.); sharida@upm.edu.my (S.F.); 3Department of Linguistics and Human Sciences, Begum Nusrat Bhutto Women, University Sukkur, Sindh 65200, Pakistan; 4Department of Human Anatomy, Faculty of Medicine and Health Sciences, Universiti Putra Malaysia, Serdang 43400, Malaysia; 5Department of Pharmacy, National University of Singapore, 119078 Singapore; phapg@nus.edu.sg

**Keywords:** coated and drug-loaded iron oxide nanoparticles, nanodelivery, surface modification, coating, polyethylene glycol

## Abstract

In the last two decades, the development of novel approaches for cancer treatment has attracted intense attention due to the growing number of patients and the inefficiency of the available current conventional treatments. In this study, superparamagnetic iron oxide nanoparticles (SPIONs) were synthesized by the co-precipitation method in an alkaline medium. Then the nanoparticles were chemically modified by coating them with polyethylene glycol (PEG) and sorafenib (SO)–zinc/aluminum layered double hydroxide (ZLDH) to improve their biocompatibility. The SPIONs and their coated and drug-loaded nanoparticles, M-PEG–SO–ZLDH are of the crystalline phase with the presence of C, O, Al, Fe, Cl, Zn in the latter, indicating the presence of the coating layers on the surface of the SPIONs. The superparamagnetic properties of the bare SPIONs were found to be reduced but retained in its coated drug delivery nanoparticles, M-PEG–SO–ZLDH. The latter has an average particle size of 16 nm and the release of the drug from it was found to be governed by the pseudo-second-order kinetic. The cytotoxicity and biocompatibility evaluation of the drug-loaded magnetic nanoparticles using 3T3 and HepG2 cells using the diphenyltetrazolium bromide (MTT) assays shows that the synthesized nanoparticles were less toxic than the pure drug. This preliminary study indicates that the prepared nanoparticles are suitable to be used for the drug delivery system.

## 1. Introduction

Cancer is a lethal disease that endangers human life and many people are suffering from it. Cancer therapy is one of the main challenges in the development of medical science for cancer treatment [1]. Since the conventional methods in cancer therapy, such as surgery and radiotherapy, are not effective in the treatment of some cancers, chemotherapy can be considered a good approach for cancer treatment [2]. Sorafenib is an important anti-liver cancer drug. However, its clinical efficiency is limited due to drug and cell resistivity [3,4].

Various methods have been studied to reduce side effects and improve the efficiency of chemotherapy. The application of nanotechnology in medical therapeutics to identify and treat diseases such as cancer is promising, and this area of research is often referred to as nanomedicine [5,6]. It has been proved that nanomedicine is rapidly developing due to its advantages to overcome limitations of the traditional drug delivery system such as non-biodegradability and poor water solubility of the drug as well as low therapeutic indicators. Additionally, nanomedicine enhances treatment efficiency and drug absorption [7], elevating persistence and decreasing the side effects [8,9].

In the last two decades, the great potential of superparamagnetic iron oxide nanoparticles (SPIONs) with their remarkable chemical and physical properties, such as nanoscale structure, magnetic characteristics, relative chemical stability, and biological compatibility, has been proved for nanomedicine applications [10]. SPIONs are of great importance for researchers in a wide range of applications including for cancer therapy, biotechnology, and drug delivery [11,12].

One of the important properties of SPIONs is the ability to control their properties, which can be tailor-made for their applications [13]. In addition, they have magnetic properties that can load and transfer the drug, and this has a great potential for the treatment of diseases, particularly cancer. The findings of the superparamagnetic iron oxide nanoparticles from recent research have shown that these nanoparticles have no immediate or long-term toxicity effects on the body [14].

SPIONs can be synthesized by various methods such as co-precipitation, microemulsion, thermal decomposition, sol–gel, etc. [15]. These methods, however, have some drawbacks such as high-temperature synthesis, high cost, and preparation of complex solutions [16]. The co-precipitation method is the simplest and the most convenient approach for synthesizing SPIONs, where an aqueous solution of iron (II) and iron (III) can be used. In addition, it does not produce toxic particles, can be prevented from aggregation using a coating agent, and a narrow particle size distribution smaller than 20 nm can be obtained.

The concept of targeted drug delivery is of great importance in cancer treatment. On the one hand, drug transfer to the tumor cells (targeting), increasing the cytotoxicity of the anticancer drug and killing cancer cells with the least impact on the healthy cells is challenging [17]. On the other hand, there is a need to minimize drug-related challenges such as low solubility, environmental degradation, and a short half-life cycle [18]. The targeted delivery of chemotherapeutic agents to cancer cells by SPIONs has been studied and has showed promising results that anticancer drug loading on the surface of SPIONs can enhance drug performance and quick removal from the bloodstream [19,20]. Nonetheless, the chemical instability of SPIONs and the agglomeration of particles are challenging [21]. Thus, the development of effective approaches for the enhancement of the chemical stability of SPIONs is of great importance. To tackle this, the surface of the magnetic nanoparticle can be covered with an impermeable protective layer or the coating shell [22].

Polymeric coating using polyvinyl alcohol (PVA), polyethylene glycol (PEG), chitosan, etc., preserves the core and its magnetic properties, which leads to the medical functionalization of the nanoparticles. PEG is one of the commonly used polymers in drug delivery systems due to its non-toxicity, non-antigenicity, and water solubility.

The PEG coating on SPIONs (M-PEG) is known as one of the important carriers in the biomedical field due to its reasonable saturation magnetization and high surface area [23,24]. In addition, the polymeric shell prevents the agglomeration of nanoparticles, increases stability and biocompatibility, and decreases biological toxicity and side effects. Liang et al. have used nanoparticles coated with polyethylene glycol to deliver doxorubicin [25] or polyarabic acid for the delivery of anticancer drugs [26]. The bond between the polymer and the surface of iron oxide is schematically depicted in Figure 1 [27]. The presence of a hydrophilic, PEG coating on the surface of SPIONs has a vast number of biological advantages, such as increasing the shelf life of the drug in the body, slowing down the breakdown of the nanocarriers by metabolic enzymes or their elimination by the immune system, and preventing nanoparticle removal [28].

Furthermore, the layered double hydroxide (LDH) can be used as the host system for drugs [29]. The superior characteristics of the LDH such as good biocompatibility, stability, and availability can be exploited for the application of LDHs in the drug delivery systems and controlled drug release [30,31,32].

Generally, targeted drug delivery using magnetic nanoparticles is achieved by applying an external magnetic field [33]. In addition, the concentration of drug-loaded magnetic nanoparticles in the region of the tumor can be increased without entering natural tissues. With the destruction of the nanoparticle and the release of the drug at tumor sites, targeted drug delivery can be achieved [34,35,36,37].

The hydrodynamic size of drug-loaded magnetic nanoparticles plays a vital role in concealing them and preventing rapid removal in the bloodstream. Particles of larger size would be easily trapped in the liver [38], whereas smaller particles are more useful for biomedical purposes [39]. Nanoparticle dispersion increases with the use of two layers of coating and leads to the formation of nearly spherical particles with an average particle size of 20 nm diameter [40].

In this research work, magnetite SPIONs were synthesized by the co-precipitation method, and then they were coated with PEG, followed by another layer, Zn/Al LDH loaded with an anti-cancer drug, sorafenib, for the formation of M-PEG–SO–ZLDH nanoparticles. Then their physico-chemical properties were characterized by different techniques. The purpose of this research work was to synthesize and characterize novel two-layered drug-loaded magnetic nanoparticles as a nanocarrier for targeted drug delivery to cancer cells using an external magnetic field.

## 2. Methods and Methods

### 2.1. Materials

Ferric chloride hexahydrate (FeCl_3_.6H_2_O) and ferrous chloride tetrahydrate (FeCl_2_.4H_2_O) (99% purity) and ammonia solution (25%) were used as starting materials in this work. They were purchased from Merck (Darmstadt, Germany). The materials to coat the surface of the SPIONs were polyethylene glycol (98%, average MW 6000) and an anticancer drug, sorafenib with purity 98.5%, which were purchased from Acros Organics company, Fair Lawn, NJ, USA, and Xi’an Yiyang Bio-Tech company, Beijing, China, respectively. The raw materials used to synthesize zinc/aluminium LDHs were zinc nitrate and aluminum nitrate (purity: 99%), which were provided from ChemAR company in Kielce, Świętokrzyskie, Poland, and dimethyl sulfoxide, used as a solvent, was obtained from Sigma Aldrich company in St. Louis, MA, USA. Ethanol and methanol solutions of the standard of high-purity HPLC grade were purchased from Sigma Aldrich company (Taufkirchen, Germany).

### 2.2. Experimental Setup

The SPIONs were synthesized by the chemical co-precipitation method of iron chloride salts in ammonia solution [19] and a layer of polyethylene glycol was formed on the surface. For this purpose, a certain amount of divalent and trivalent iron chloride salts were dissolved in distilled water with a molar ratio of 1:2 (Fe^+2^/Fe^+3^ = 2:1) under a nitrogen atmosphere (oxygen-free to prevent oxidation) and at room temperature and stirred with a mechanical stirrer for a few minutes. The chemical co-precipitation reaction was done by adding a 6 mL ammonium solution, then, the magnetite nanoparticles formed were separated using a magnet and washed three times by deionized water. Then, the dissolution of 2% polyethylene glycol led to the black precipitation by autoclaving at 150 °C for one day. The deionised water (DI) water was used for removing residual materials and their impurities followed by drying at 70 °C. In the next step, the basic solution prepared by the dissolution of sorafenib in dimethyl sulfoxide was mixed with the as-synthesized black powders under stirring using a magnetic stirrer for one day. The co-precipitation method was used to prepare Zn/Al LDH. For this purpose, the salt solution of Al(NO_3_)_3_.9H_2_O and Zn(NO_3_)_2_.6H_2_O with a Zn/Al molar ratio of 4 was prepared using a four-mouth flask under a nitrogen gas environment. Then, NaOH solution was added into the solution dropwise at a rate of 1 drop per minute. This NaOH solution was continuously added until the solution reached pH = 7. Then, the Zn/Al LDH was mixed with the prepared nanoparticles. The resulting sample was collected followed by centrifugation upon the completion of the previous process. Finally, the samples were washed three times and oven-dried.

### 2.3. Instrumentation

The existence of the coating agents and the drug was determined by Fourier-transform infrared (FTIR) spectroscopy, (Thermo Nicolet 6700, with high-resolution standard 0.09 cm^−1^, Madison, WI, USA) in the range of 500–4000 cm^−1^. The particle size distribution and morphology of the nanoparticles were analyzed by high-resolution scanning transmission electron microscopy (HRTEM, Hitachi H-7100, Tokyo, Japan). The magnetic properties of the nanoparticles were determined using a vibrating sample magnetometer (VSM) instrument (Lakeshore 7404, Westerville, OH, USA). The crystalline structure of the nanoparticles was determined by X-ray diffraction (XRD 6000, CuK_α_ radiation, 40 kV, 30 mA, Shimadzu, Japan), in the range of 2°–80°). It should be noted that this range of scans covers all the major reflections of the superparamagnetic iron oxide nanoparticles, drug, and other coating agents. Measurement of the drug loading was determined by high-performance liquid chromatography technique (HPLC, Alliance e2695, Milford, MA, USA). The thermal behavior of the synthesized nanoparticles was studied using thermogravimetric/differential thermogravimetric (TGA/DTG) analyses. A heating rate of 10 °C/min was used from 20 to 1000 °C. The microstructure of the synthesized samples was investigated by a field emission scanning electron microscope (FESEM, NOVA NANOSEM 230 model, Denton, TX, USA). Due to the nonconductivity of the samples, they were coated with a layer of gold before the test to improve the quality of the images. In order to evaluate the drug loading and release, UV–visible spectroscopy was used, using a PerkinElmer, Lambda 35 from the USA. The particle size and the size distribution of the nanoparticles were studied using a dynamic light scattering method (DLS), Malvern, NanoS, U.K.

### 2.4. Methylthiazol Tetrazolium Cell Viability Assay

For cell culture test, Roswell Park Memorial Institute (RPMI, 1640 medium), antibiotics containing penicillin and streptomycin, trypsin, and M-EDTA were purchase from Nacalai Tesque (Kyoto, Japan). The MTT compound (3-[4,5 dimethylthialzol-2-yl]-2,5 diphenyltetrazolium bromide) and fetal bovine serum (FBS) were bought from Sigma Aldrich, St. Louis, MO, USA. The normal human fibroblast (3T3) and human hepatocellular carcinoma cells (HepG2) were provided from the cell bank at the National Center of Genetic Engineering and Biotechnology. Deionized water was used in all the experiments.

MTT analysis is a standard color test that shows the cell viability and cytotoxicity level of nanoparticles [23]. All the solutions were prepared by dissolving the compound in 1:1 of 0.1% DMSO and RPMI, and the cells were treated with pristine sorafenib (SO), M-PEG–ZLDH (nanocarriers), M-PEG–SO–ZLDH (the drug delivery nanoparticles). In order to evaluate the toxicity of nanoparticles in the vicinity of two different 3T3 and HepG2 cell lines, in the beginning, the cells were grown using RPMI and the cell culture medium was supplemented with 10% FBS along with 1% antibiotics containing 10,000 units/mL penicillin and 10,000 μg/mL streptomycin. Then the cells were incubated in humidified 5% CO_2_ 95% room air at 37 °C. Layers of the cell were harvested using 0.25% trypsin and 1 m M-EDTA, 1.0 × 10^4^ cell numbers were transferred and seeded in 96-well tissue culture plates and placed in an incubator for 24 h to attach, and 80% confluence was achieved for treatment. The diluted mixture in the same medium was obtained in various concentrations (from 1.25 to 100 μg/mL). The tested nanoparticles were added to each of the well plates until a final volume of 100 μL and the well plates were kept in an incubator for another 24 h. After that, 10 μL of 5 mg/mL MTT was added to each well plate, and they were kept in an incubator for 3 h before aspiration. Then, the purple formazan salt was dissolved by the addition of 100 μL of DMSO to each well plate. After 15 min, the light absorbance of the purple color solution was measured at 570 nm wavelength by a microplate reader (Biotek LE800, Winooski, VT, USA).

## 3. Results and Discussions

### 3.1. Powder X-Ray Diffraction

Figure 2 shows the X-ray diffraction (XRD) patterns of the synthesized nanoparticles. As can be observed from Figure 2A, the characteristic peaks related to the synthesized SPIONs were observed at 2θ = 35°, 41.3°, 50.4°, 62.9°, 67.2°, and 74.1° with corresponding planes of (220), (311), (400), (511), and (440), respectively, in which all the reflections match with the Joint Committee on Powder Diffraction Standards (JCPDS) reference no. 85–1436 [41]. The position of these reflections indicated that the SPIONs were of single phase and confirmed the inverse spinel structure with a chemical formula of Fe_3_O_4_. The absence of extra peaks in the XRD pattern indicates the high purity of the synthesized sample. Since the X-ray diffraction pattern of Fe_3_O_4_ and Fe_2_O_3_ phases are similar to each other, it seems that the synthesized samples could be a mixture of magnetite and maghemite phases [42].

The observed main diffraction peaks at the 2θ = 11.5°, 23.2°, and 34.8° and 2θ positions between 10° and 35° together with 2θ = 19.3° and 23.5° in Figure 2B–D are related to the Zn/Al layered double hydroxides, sorafenib, and polyethylene glycol, respectively. All peaks with high intensity were present in the final sample. The XRD pattern of the final sample (Figure 2E) shows that SPIONs were the dominant phase in the sample. The characteristic of the diffraction peak in the XRD patterns of the naked sample indicates that there are no impurities in the samples, indicating the pure phase of the sample. As can be observed, by creating a surface layer on the nanoparticles, the XRD pattern of the SPIONs was different and the peaks’ intensity was reduced, and the peak widths were also increased. This indicates the existence of new coated materials on the surface of the SPIONs, as well as the reduction in the peak intensity of sorafenib and ZLDH. This is because when the drug was replaced with a nitrate anion in between the LDH interlayers, peaks were displaced to lower 2θ angles. Based on the high saturation magnetization of the samples, and their completely black color after the reaction had taken place when the experiment was performed under inert nitrogen gas, this presumably indicates that there is a negligible non-magnetic phase in the final sample.

### 3.2. Infrared Spectroscopy

Fourier-transform infrared spectroscopy (FTIR) analysis is a suitable method to determine the functional groups that are present in organic compounds of the sample. Thus, FTIR analysis was performed on the synthesized nanoparticles to determine functional groups and the existing bonds in the samples. For this purpose, the FTIR spectra of the nanoparticles were obtained using the KBr-pellet method and the final spectra were obtained in the range of 500–4000 cm^−1^.

The FTIR spectra and their corresponding wavenumbers related to the superparamagnetic iron oxide nanoparticles (Figure 3A) show a broad band at 3318 cm^−1^ that linked to a strong vibrational band resulting from the hydroxyl groups (OH) and OH….OH vibrational bands. This band was shifted to lower wavenumbers in the drug-loaded magnetic nanoparticles (3277 cm^−1^), indicating that the hydrocarbon chains surrounded the surface of SPIONs. For the FTIR spectra of pure polymer, the band at 2881 cm^−1^ is attributed to the symmetric and asymmetric stretching vibration of C–H bond in the polymer, which was shifted to 2925 cm^−1^ in the drug-loaded magnetic nanoparticles. The bands at 2361 and 1712 cm^−1^ can be attributed to the C=O and C–O bond stretching, respectively. The bands at 1630–1670 cm^−1^ are ascribed to the stretching vibrations of C–H from the polymer and the drug, and they remained in the synthesized M-PEG–SO–ZLDH. The band related to water (H_2_O) adsorption is observed at around 1619 cm^−1^. A band at 1346 cm^−1^ shows the presence of nitrate groups in the structure. The absence of a band at around 1400 cm^−1^ in the drug-loaded magnetic nanoparticles indicates that all the nitrate anions were replaced with the drug, indicating that the resulting nanoparticles do not contain nitrate anions [41].

As can be observed, bands below 1000 cm^−1^ are usually due to ferrites. Therefore, the band at 531 cm^−1^ is attributed to the Fe–O vibrational band with the tetrahedral condition, indicating the formation of the spinel structure. The appearance of these bands indicates that the main phase of the synthesized drug-loaded magnetic nanoparticles is superparamagnetic iron oxide nanoparticles. In the FTIR spectra, the vibrational band for the hydroxy groups was also observed, but this is due to the high moisture absorption on the sample [43,44].

As can be observed in Figure 3, there is no significant difference between the FTIR spectrum of polyethylene glycol and M-PEG–SO–ZLDH. Furthermore, the high absorption is due to the C–H bonds associated with PEG in the final sample, which illustrated that the surface of the magnetite nanoparticles was fully covered by the polymer and/or other coating agents. It is noteworthy that the FTIR analysis confirmed the presence of SPIONs in the final sample.

### 3.3. Magnetic Properties

Figure 4 shows the hysteresis curve of SPIONs and their drug-loaded product, M-PEG–SO–ZLDH nanoparticles obtained using a vibrating sample magnetometer (VSM) at room temperature and under fields of −20,000 to 20,000 Oersted, showing their values for saturation magnetization (M_s_), remnant magnetization (M_r_), and their high coercivity (H_ci_); these are summarized in Table 1.

As can be seen, the magnetic curve of the nanoparticles passes through the origin. In addition, the coercive field and remanent magnetization are not observed. This indicates that the synthesized nanoparticles have superparamagnetic properties at room temperature. In other words, SPIONs and drug-loaded magnetic nanoparticles became magnetized upon the application of the magnetic field. However, this magnetism is not permanent and is undone by the elimination of the magnetic field [45].

The maximum magnetic saturation for the SPIONs was found to be 73 emu/g compared to 30 emu/g for the M-PEG–SO–ZLDH nanoparticles. A very low value of H_ci_ for the synthesized magnetic nanoparticles confirmed their mono-regional characteristic. It was clear that the increase of the coating agents on the SPIONs resulted in the value of saturation magnetization also decreasing. As can be observed, the hysteresis curves for both the samples are S shaped, since the synthesized sample could be a mixture of Fe_3_O_4_ with relatively high magnetic saturation and a minor amount of non-magnetic materials [46]. This is presumably due to the decline in the saturation magnetization. The higher saturation magnetization (M_s_) value of the Fe_3_O_4_ nanoparticles compared to that of their drug-loaded magnetic nanoparticles could be attributed to the presence of polymer and other materials and also to the remaining impurities on the surface of Fe_3_O_4_ nanoparticles.

### 3.4. Thermal Analysis

The thermogravimetric/differential thermogravimetric analyses (TGA/DTG) are the simplest thermal analyses that can be used to investigate the effect of temperature on the sample, thermal stability, and the measurement of sample weight loss during the heating process. The TGA/DTG thermograms at 25–1000 °C are shown in Figure 5.

As can be seen, the thermal stability of drug-loaded magnetic nanoparticles was considerably increased up to 200 °C, and there was no weight loss below that temperature. However, at 800 °C, 30% weight loss was observed. The TGA/DTG thermograms of the drug-loaded magnetic nanoparticles (Figure 5E) show that the first mass loss was at 209 °C, due to the release of the residual water from the sample. The second weight loss curve (2.7%) is due to the carbonization, i.e., the conversion of organic content and also the evaporation of the outer layer. The third weight loss at 290–500 °C is believed to be due to the decomposition of polyethylene glycol and the degradation of PEG chains, followed by the complete degradation of polymer chains. The last, fourth weight loss is due to the interlayer anion decomposition and the collapse of the layered structure. At temperatures above 1000 °C, the 2D layered structure of LDH is no longer maintained, and new oxide phases are formed.

These results show that the coated nanoparticles have higher thermal stability compared to their non-coated counterparts. Additionally, the different thermal behavior of these samples confirms the successful surface modification of SPIONs by coating them with coating agents. Previous work has shown that the TGA/DTG analysis of iron oxide nanoparticles showed 5% weight loss due to the presence of moisture in the sample, which is close to the 6.8% weight loss in our present study. In addition, in the above-mentioned study, the stability of the coated nanoparticles was increased up to 250 °C and the final decomposition temperature was reported to be 700 °C, which is in agreement with our present study [47].

### 3.5. Surface Morphology and Particle Size

In order to perform the morphological analysis of the produced samples, field emission scanning electron microscopy (FESEM) analysis with 200,000× magnification was conducted for the samples. Figure 6 illustrates the apparent condition of coating agents on the SPIONs. The FESEM images show that the coated SPIONs have near-spherical particles and exist in the form of chain-like masses with almost similar particle size. Furthermore, the nanoparticles have the core–shell structure, where the SPIONs as core are layered by the shell, the coating agents. Particles are attached to each other due to the adhesive characteristic of the polymer and other bonds that are present on the surface of SPIONs, therefore the existence of prominent agglomerates was unavoidable. In addition, the nanoparticles have a high ratio of surface area to volume due to their small size. This characteristic allows multiple ligands to attach to the surface and increases their inclination to form agglomerations. As a result, it was difficult to obtain completely well-separated nanoparticles and a stable system, and therefore aggregation of particles is usually unavoidable. This depends on several factors such as interactions between the SPIONs themselves and the chemistry of the coating agents, as well as the thickness of the coating layers.

The FESEM micrograph displayed in Figure 6 provides a summary of the EDX analysis. Based on the table, the presence of the five most common elements; oxygen, iron, carbon, aluminum, and zinc can be observed in the final coated sample. The atomic percentage of carbon (from the carbon of the polymer structure), iron (from the SPIONs), oxygen, aluminum, and zinc (from the Zn/Al LDH) is 10.2%, 1.7%, 33%, 46.9%, and 7.8%, respectively.

In order to determine the nanoparticle size distribution, dynamic light scattering (DLS) analysis was adopted. This analysis was used to investigate the particle size distribution and the average particle size of the samples. Based on Figure 7, the average particle size of the coated final sample is smaller than the SPIONs, a non-coated sample. The result of the zeta sizer analysis using the DLS method shows that 70% of the synthesized SPIONs have a monomodal particle size distribution with a diameter of 190 nm (Figure 7A) with a narrow size distribution. After the surface modification of SPIONs by coating them, 62% of the nanoparticles also had a narrow particle size distribution, between 68 and 122 nm with a diameter of 91 nm. The reason for this is that polyethylene glycol is a polymer with a long chain, which could reduce the agglomeration of the nanoparticles. In addition, the increase in the number of coating agents in the final coated nanoparticles reduces the size of the nanoparticles, and the growth of the nanoparticles is restricted during the germination. This indicates that the presence of fewer agglomerates in the drug-loaded magnetic nanoparticles is due to the presence of the coating agents. A previous study has found that the average diameter for SPIONs measured by the DLS analysis was 218 nm, compared to 190 nm in our present study [47].

It is worth noting that the nanometer-size regime and suitable particle size distribution could play a positive role in the stability of the suspension of the drug-loaded magnetic nanoparticles, and this makes the nanoparticles suitable for targeted drug delivery applications.

### 3.6. Particle Size Distribution and Morphology

In order to look at the particle size and crystal structure of the SPIONs coated with polyethene glycol–sorafenib–Zn/Al LDH, high-resolution transmission electron microscopy (HRTEM) analysis was used. The HRTEM images of the synthesized samples are shown in Figure 8.

The particle size distribution was determined using image analysis software (Image J) from 110 particles selected randomly. As shown in the figure, the estimated drug-loaded magnetic nanoparticles’ mean diameter is around 16 nm with a narrow particle size distribution, which is ideal for the drug nanodelivery application. It could be inferred that the nanoparticles were of single crystal, and since the superparamagnetic iron oxide particles become single domain and monopolar in the sizes below 50 nm, the drug-loaded magnetic nanoparticles were also of single domain and monopolar. Additionally, the synthesized nanoparticles have a spherical morphology and near-uniform structures.

### 3.7. Determination of Percentage of Sorafenib in the Drug-Loaded Magnetic Nanoparticles

To determine the percentage of anticancer drug loaded on the magnetic nanoparticles, first, a standard curve was prepared using various concentrations (0, 25, 50, 100, 150, and 200 ppm) of sorafenib. Then the drug-loaded magnetic nanoparticle solution was put in methanol and then exposed to ultrasound for 1 h. The percentage of sorafenib in the drug-loaded magnetic nanoparticles was measured by high-performance liquid chromatography (HPLC) with a photodiode detector array and with a mobile phase containing acetonitrile and phosphate-buffered solution at pH = 7 with a flow rate of 1.5 mL/min, with column C18 Agilent (3.9 × 150 mm). The percentage of drug loading for sorafenib in M-PEG–SO–ZLDH compound was found to be 59% (Figure 9A) using the standard curve, indicating the successful internalization of the drug in the coating layer of the nanoparticles.

### 3.8. In Vitro Drug Release Study

The release profiles of the physical mixture of the anticancer drug and the drug-loaded magnetic nanoparticles were studied in PBS media at pH = 7.4 (similar to human plasma) and 4.8 (similar to cancer area) and the percentages of drug released were determined using ultraviolet–visible spectroscopy, as shown in Figure 10.

Ssorafenib’s release profiles in the physical mixture and its drug-loaded magnetic nanoparticles show that the maximum percentage of drug release from the beginning to the first 40 min was 82% in the neutral buffer and 91% in the acidic buffer. This means that the drug release was rapid at first and gradually became slow and leveled off thereafter.

Polyethylene glycol is a biocompatible compound such that the encapsulation of sorafenib in it and other coating agents leads to a significant decline in the adverse effects and intrinsic toxicity of sorafenib. In addition, polyethylene glycol is a biodegradable compound with high permeability and the presence of a hydrophilic part beside the hydrophobic part, the drug (sorafenib) causes the fast migration of sorafenib molecules to the surface and rapid initial release. Sorafenib was markedly released in acidic pH due to acid-accelerated hydrolysis of the PEG polymer. On the other hand, low electrostatic attraction between the sorafenib anions and the superparamagnetic iron oxide–polyethylene glycol in the physical mixture demonstrated no sustained-release effects.

The study of the drug release pattern from the drug-loaded magnetic nanoparticles indicates that the drug release is slow as expected, indicating that the presence of the coating agents played an effective role to slow down the drug release. This is because the drug needed more time to pass through the barrier, the surface layers, and the release was found to be around 91% and 85% within 5700 min for pH = 4.8 and pH = 7.4, respectively (Figure 11).

The slow behavior of sorafenib release from the drug-loaded magnetic nanoparticles compared to the physical mixture, shows that drug-loaded magnetic nanoparticles allow the controlled release of the drug in the drug delivery system and may benefit the treatment of cancer. Furthermore, the difference in the release mechanism of sorafenib from the drug-loaded magnetic nanoparticles (between pH 7.4 and pH 4.8) is ascribed to greater stability of the nanoparticles at the acidic pH, where the release would have occurred via the ion-exchange mechanism. The release in the second 300 min became much slower and more sustained compared to that in the initial 300 min probably due to the release of the free, uncoated drug.

### 3.9. Kinetics Release of Sorafenib from Its Drug-Loaded Magnetic Nanoparticles

The kinetic release studies can be used to describe the change in the concentration over time to investigate factors that affect the reaction rate. For this, first-order Equation (1), pseudo-second-order Equation (2), and parabolic diffusion Equation (3) release kinetic models were used, and the amount of the drug released into the PBS solutions at pH 4.8 and pH 7.4 at the preset time was determined using the UV/Vis spectroscopy, and the results are shown in Figure 12.
In (q_e_ − q_t_) = In q_e_ − kt(1)
t/q_t_ = 1/k_2_ q_e_^2^ + t/q_e_(2)
(1 − M_t_/M_0_)/t = k_t_^−0.5^ + b(3)
where q_e_ is the amount adsorbed drug at equilibrium time, q_t_ is the adsorbed amount at any time t, k is the corresponding rate constant, M_o_ is the amount of drug remaining at release time 0 and M_t_ is the drug content remaining at release time t [38].

Using these three kinetic models, the pseudo-second-order was found to give the best fit, and therefore it is the most satisfactory kinetic model that can describe the release of the drug from the M-PEG–SO–ZLDH nanoparticles.

Figure 12 displays plots of t/q_t_ against time for the pseudo-second-order kinetic model and other models used in this study. It was observed that the pseudo-second-order kinetic model gave a straight line. The resulting correlation coefficient (R^2^), percentage of saturation release, rate constant (k), and half time (t_1/2_) are summarized in Table 2. The release rate constant was found to be 4.31 × 10^−3^ and 4.32 × 10^−3^ mg/min at pH 7.4 and pH 4.8, respectively, and the correlation coefficient was 0.9986 and 0.9992 at pH 7.4 and 4.8, respectively.

### 3.10. In Vitro Bioassay

Cell viability assay was conducted to analyze the toxicity level of the nanoparticles to cell lines. Two types of cells were used for the cytotoxicity assays; normal human fibroblasts (3T3) and human hepatocellular carcinoma cells (HepG2), which were purchased from American Type Culture Collection (ATCC) (Manassas, VA, USA). All the cells were grown using Dulbecco’s modified Eagle medium (DMEM) (Nacalai Tesque, Kyoto, Japan) supplemented with 10% fetal bovine albumin (Sigma Aldrich, St. Louis, MO, USA), 1% antibiotics containing 10,000 units/mL penicillin, and 10,000 μg/mL streptomycin (Nacalai Tesque, Kyoto, Japan). Cells were maintained and incubated in humidified 5% carbon dioxide at 37 °C. Cell layers were harvested using 0.25% trypsin/1 m M-EDTA (Nacalai Tesque, Kyoto, Japan). This was followed by seeding in 96-well tissue culture plates at 1.0 × 10^4^ cells/well for 24 h in an incubator to attach and 80% confluence was attained for treatment. A methylthiazol tetrazolium (MTT)-based assay was carried out to determine the cell viability and cytotoxicity. Cells were treated with Fe–O, M-PEG, M-PEG–ZLDH (the nanocarriers), pristine sorafenib, and M-PEG–SO–ZLDH, where stock solutions were prepared by dissolving the compound in 1:1 of dimethyl sulfoxide (0.1%) and DMEM. Then, the mixture was further diluted in the same media to produce various final concentrations ranging from 1.25 to 100 μg/mL. Once the cells were attached to the respective wells after 24 h, the tested compounds were added until a final volume of 100 μL per well was obtained. After 72 h of incubation, 10 μL of MTT solution (5 mg/mL in PBS) was added in each well and further incubated for 3 h before being aspirated. Then 100 μL of dimethyl sulfoxide was added per well in the dark at room temperature in order to dissolve the purple formazan salt. The intensity of the purple formazan solution, which reflects cell growth was subsequently measured at a wavelength of 570 nm using a microplate reader (Biotek LE800, Winooski, VT, USA).

All the cytotoxicity assays were carried out in triplicates and the standard deviations were calculated and are incorporated in the respective bar graphs. For the calculation of the half-maximal inhibitory concentration value (IC_50_), a plot of the *x*- against the *y*-axis was done and converted the x-axis values (conc.) to their log values, followed by the nonlinear regression (curve fit) under the *xy* analysis to obtain a straight line equation fit, *y* = a*x* + b, from which the regression line and then IC_50_ was calculated.

#### 3.10.1. Cell Viability Studies against Healthy Fibroblast Cells, NIH 3T3

Cell cytotoxicity studies were conducted by treating the Fe–O, M-PEG, M-PEG–ZLDH (the nanocarriers), pristine sorafenib, and M-PEG–SO–ZLDH with normal fibroblast, 3T3 cells. Various gradient concentrations of the samples were incubated for a maximum of 72 h with the 3T3 cells. Figure 13 shows the percentage cell viability of the 3T3 cells after 72 h incubation for all the samples. All of the samples show 80% of cell viability in the range of concentration for Fe–O (1.25–50 µg), M-PEG (1.25–100 µg), M-PEG–ZLDH (the nanocarriers) (1.25–6.25 µg), pristine sorafenib (1.25–25 µg), and M-PEG–SO–ZLDH (1.25–12.5 µg) and were found to be biocompatible and nontoxic after 72 h incubation. This suggests that the designed anticancer nanoparticle formulation is biocompatible with normal cells and would be very useful for targeting cancer cells without damaging/harming normal tissues. The statistics, ANOVA, revealed that no significant difference was found among the sample groups at the individual from 1.25 to 12.5 µg concentrations using ANOVA and Duncan’s multiple range test.

#### 3.10.2. Anticancer Action against Liver Cancer Cells, HepG2

The anticancer activity of Fe–O, M-PEG, M-PEG–ZLDH (the nanocarriers), pristine sorafenib, and M-PEG–SO–ZLDH samples on liver cancer cells, HepG2, was studied, and the results are shown in Figure 14. Different concentrations of the above samples were incubated with liver cancer cells, HepG2, for 72 h. The empty nanocarriers Fe–O, M-PEG, and M-PEG–ZLDH from the concentration of 1.25–25 µg did not show any inhibitory action against liver cancer cells, HepG2. The IC_50_ of the pristine sorafenib against liver cancer cells, HepG2, was found to be 21.58 μg/mL compared to 15.66 μg/mL for the M-PEG–SO–ZLDH nanoparticles. The effective IC_50_, which is the actual amount of sorafenib present in IC_50_ of the anticancer nanoparticle was calculated from the percentage of the drug loading, which is 79% for M-PEG–SO–ZLDH. This was determined using HPLC analysis. These results suggest that M-PEG–SO–ZLDH nanoparticles at 15.66 μg/mL have much better anticancer activity compared to that of the pristine sorafenib drug. The M-PEG–SO–ZLDH nanoparticles were found to be significantly different from all the other samples at concentrations of 1.25–100 μg/mL with (*P*-values of <0.5).

Statistical analysis was determined using several softwares; SPSS and ANOVA and Duncan’s multiple range test. Significant differences were found between the Fe–O, M-PEG, M-PEG–ZLDH (the nanocarriers), pristine sorafenib, and M-PEG–SO–ZLDH (the drug delivery nanoparticles). At a concentration of 12.5–100 μg/mL, the sorafenib sample was significantly different from the empty nanocarrier. The value was found to be *p* < 0.05. The samples of pristine sorafenib and M-PEG–SO–ZLDH (the nanoparticles) showed anticancer effect toward the cell line in a dose-dependent manner. The half-maximal inhibitory concentration value (IC_50_) of all the samples is given in Table 3. The IC_50_ values of the nanoparticles determined based on percentage drug loading indicate that the synthesized nanoparticles have a better anticancer effect than the drug in its free form.

## 4. Conclusions

In cancer treatment, nanodrug delivery systems have been shown to play a significant role in improving the efficiency of the treatment. In the present study, the design and synthesis of a nanocarrier for the drug delivery system aimed to have a more efficiently controlled drug release at the target site and at the same time to enhance the properties for better bioavailability of the drug. SPIONs of pure phase were successfully synthesized by the co-precipitation method followed by chemical modification using PEG and sorafenib–zinc/aluminum layered double hydroxide coating. The presence of the coating materials significantly prevented the agglomeration of the nanoparticles. Despite the presence of the coating layers on the surface of the SPIONs, the magnetic properties were maintained. The drug release from the nanoparticles was found to be governed by the pseudo-second-order kinetic, and they were biocompatible on 3T3 cells, while showing cytotoxicity toward HepG2 cancer cells. This study shows that the synthesized nanoparticles could improve the anticancer effect of the drug, sorafenib, compared to their free counterparts.

## Figures and Tables

**Figure 1 polymers-12-02716-f001:**
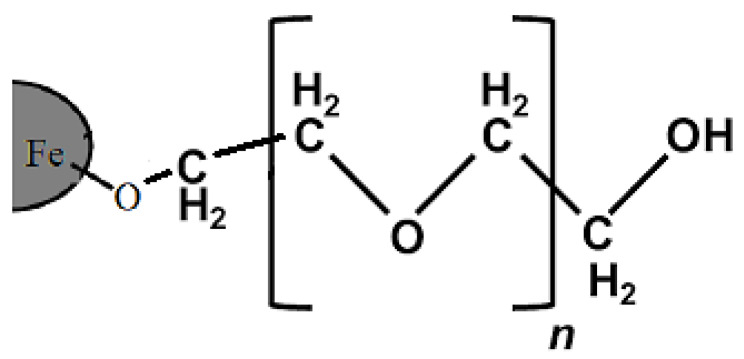
Stabilization of superparamagnetic iron oxide nanoparticles by polyethylene glycol.

**Figure 2 polymers-12-02716-f002:**
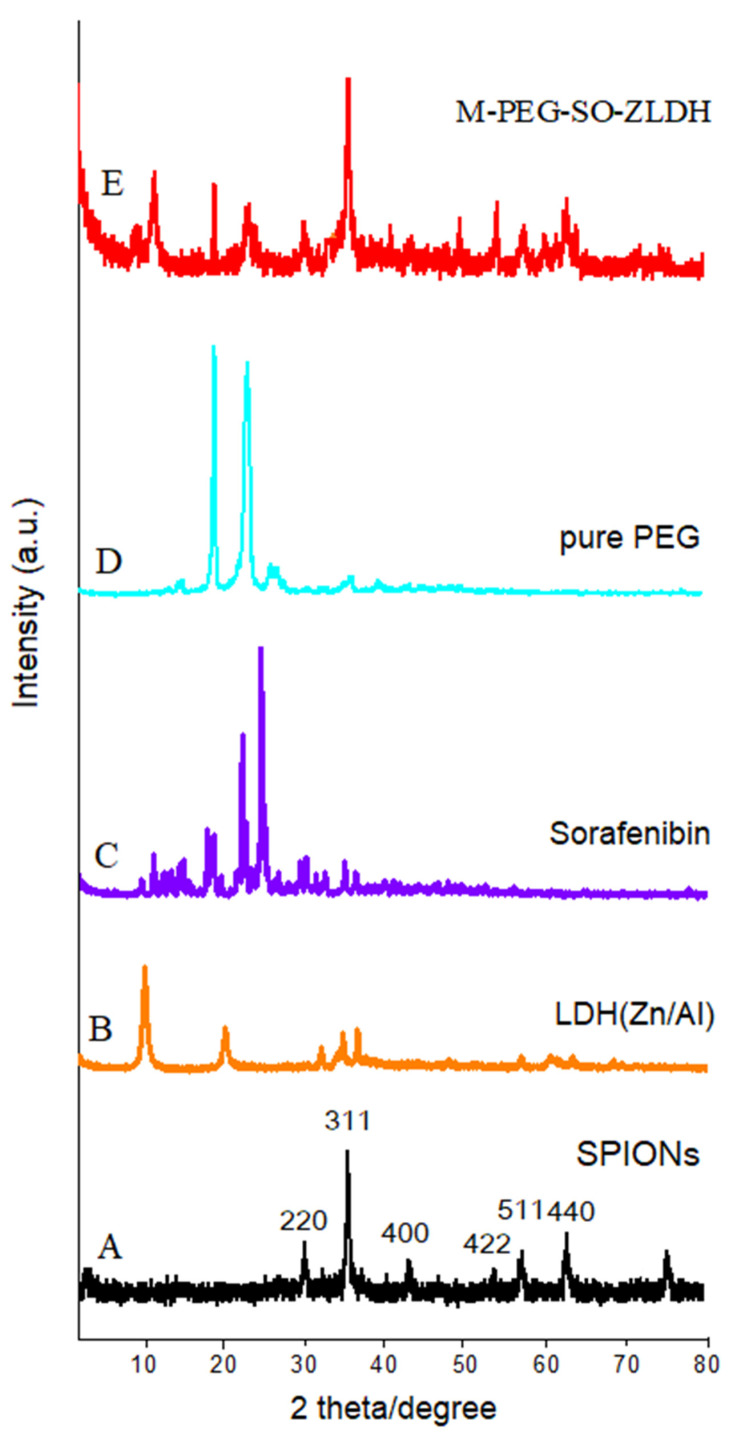
XRD patterns for (**A**) superparamagnetic iron oxide nanoparticles; (**B**) pure layered double hydroxide (LDH) (Zn/Al); (**C**) pure sorafenib; (**D**) pure PEG, (**E**) drug-loaded magnetic nanoparticles (M-PEG-SO-ZLDH).

**Figure 3 polymers-12-02716-f003:**
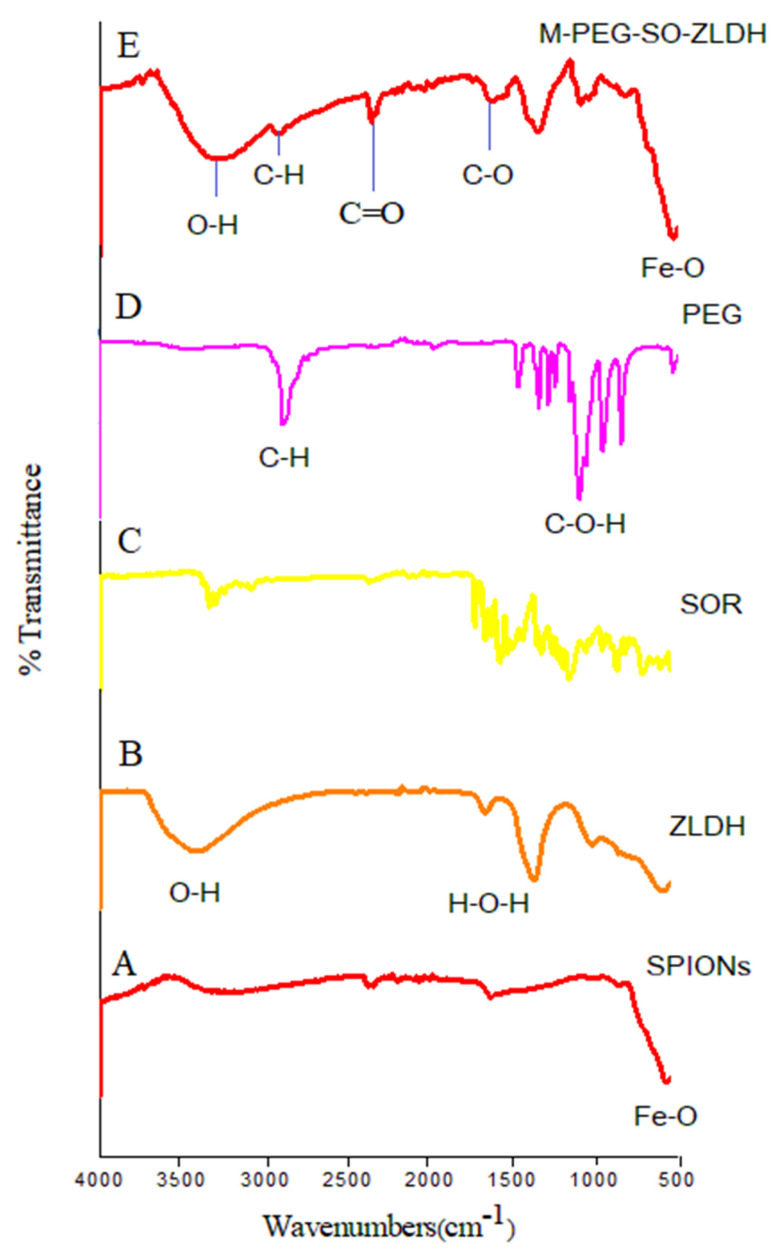
FTIR spectra for (**A**) superparamagnetic iron oxide nanoparticles; (**B**) pure ZLDH; (**C**) pure sorafenib; (**D**) pure PEG; and (**E**) drug-loaded magnetic nanoparticles (M-PEG–SO–ZLDH).

**Figure 4 polymers-12-02716-f004:**
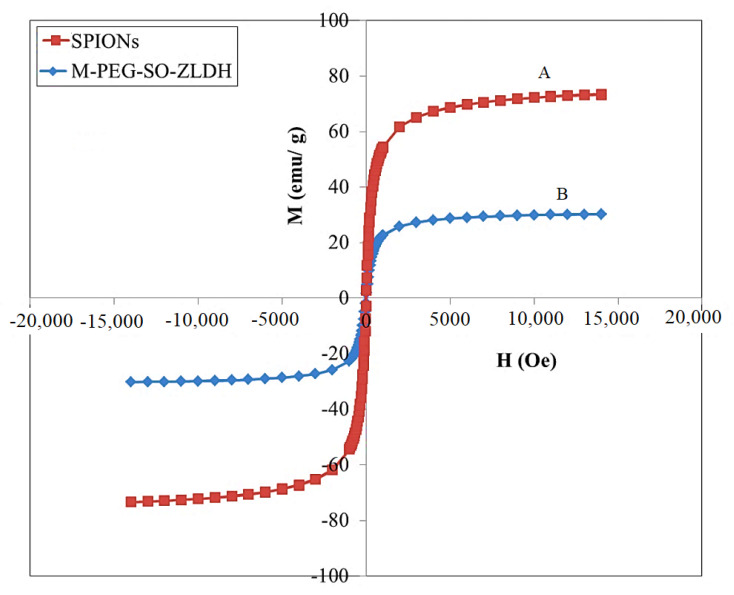
Magnetization curves of (**A**) superparamagnetic iron oxide nanoparticles and (**B**) drug-loaded magnetic nanoparticles (M-PEG–SO–ZLDH).

**Figure 5 polymers-12-02716-f005:**
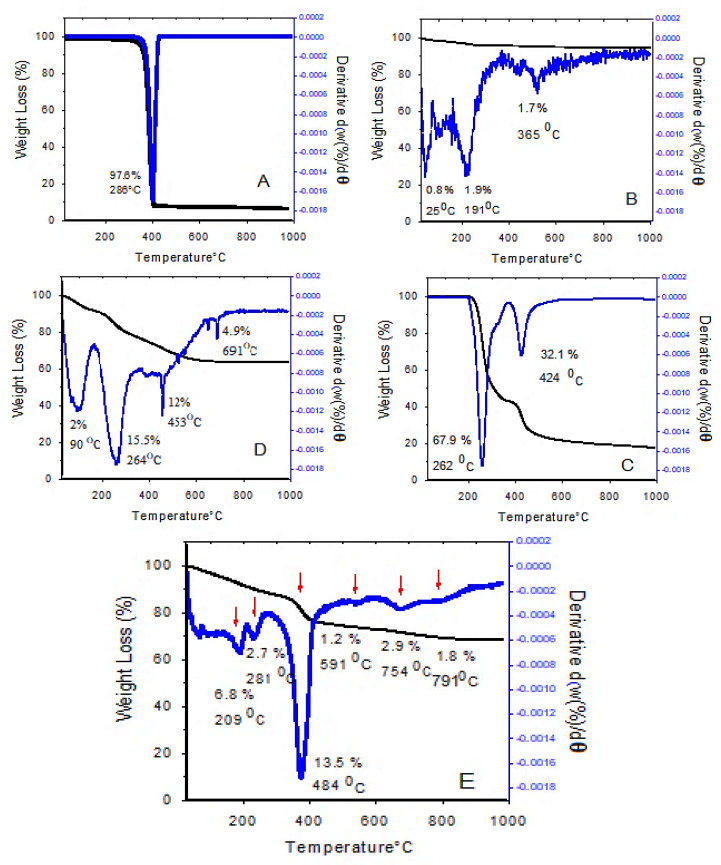
Thermogravimetric/differential thermogravimetric analyses (TGA/DTG) thermograms of (**A**) pure PEG; (**B**) superparamagnetic iron oxide nanoparticles; (**C**) pure sorafenib; (**D**) pure Zn/Al LDH; and (**E**) drug-loaded magnetic nanoparticles (M-PEG–SO–ZLDH).

**Figure 6 polymers-12-02716-f006:**
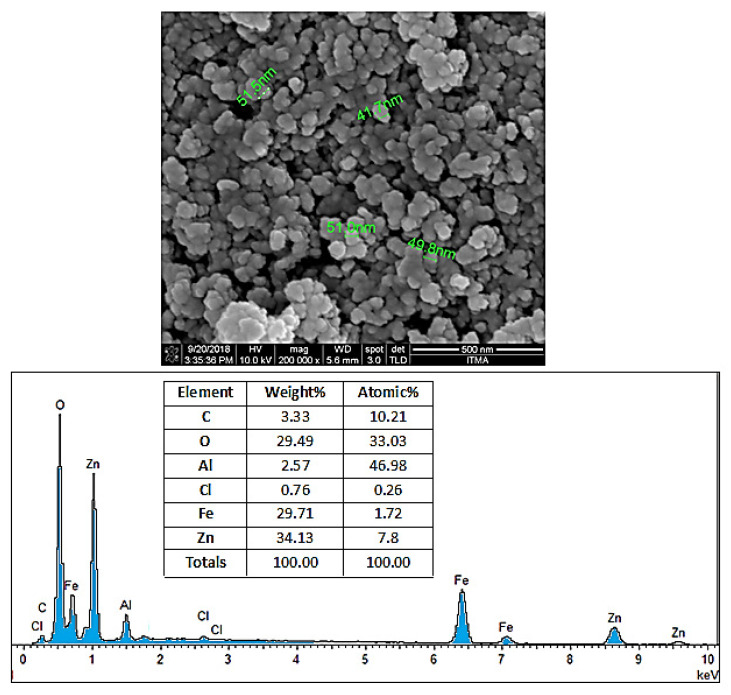
Field emission scanning electron microscope (FESEM) images of drug-loaded magnetic nanoparticles (M-PEG–SO–ZLDH) and their FESEM-EDX spectra. The specimen holder is made of aluminum, therefore a high percentage of aluminum is expected and therefore this analysis is not valid for aluminum content analysis.

**Figure 7 polymers-12-02716-f007:**
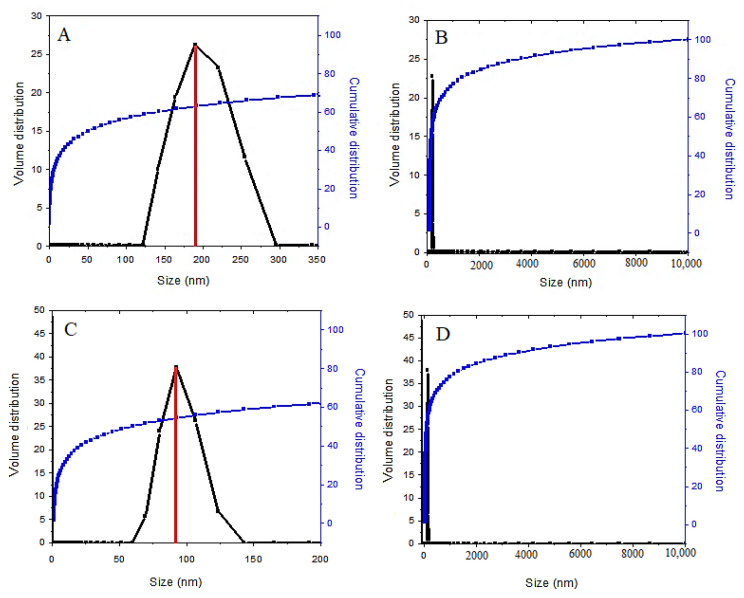
The relative (**A**) and cumulative (**B**) particle size distribution of superparamagnetic iron oxide nanoparticles; the relative (**C**) and cumulative (**D**) particle size distribution of drug-loaded magnetic nanoparticles (M-PEG–SO–ZLDH).

**Figure 8 polymers-12-02716-f008:**
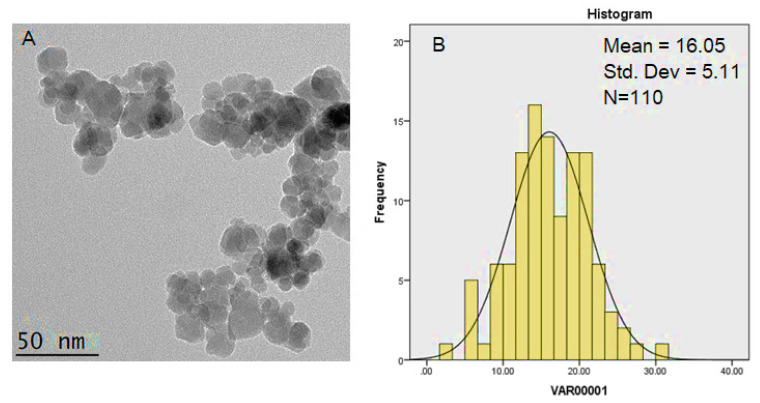
Transmission electron micrographs for (**A**) drug-loaded magnetic nanoparticles (50 nm bar); (**B**) their particle size distribution.

**Figure 9 polymers-12-02716-f009:**
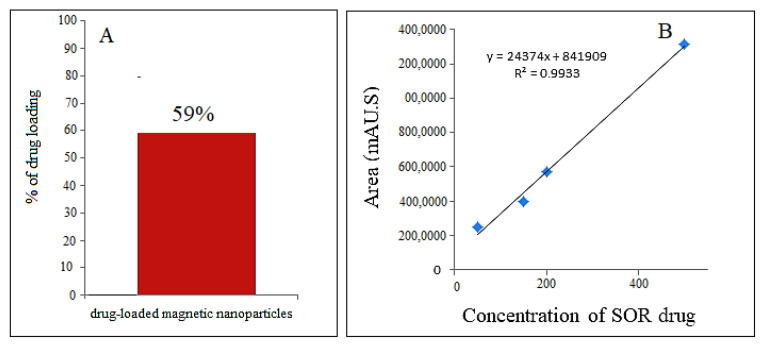
The high-performance liquid chromatography (HPLC) column chart of (**A**) drug-loaded magnetic nanoparticles; (**B**) the calibration curve.

**Figure 10 polymers-12-02716-f010:**
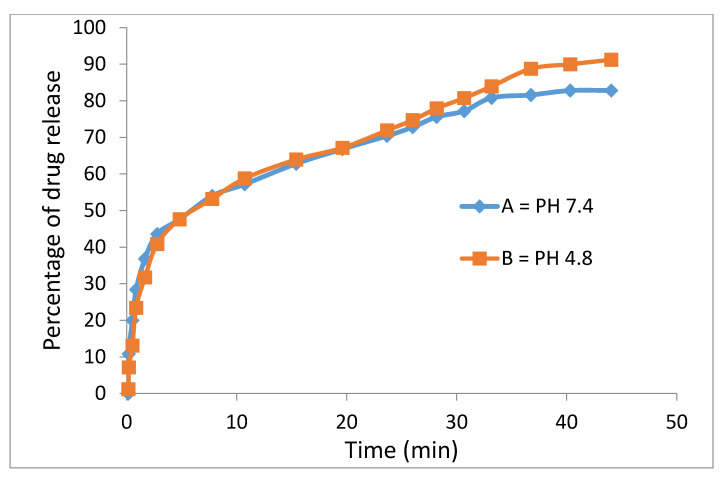
Release profiles of sorafenib from its physical mixtures in (**A**) M-PEG–SO–ZLDH in phosphate-buffered solution at pH 7.4; (**B**) M-PEG–SO–ZLDH in phosphate-buffered solution at pH 4.8.

**Figure 11 polymers-12-02716-f011:**
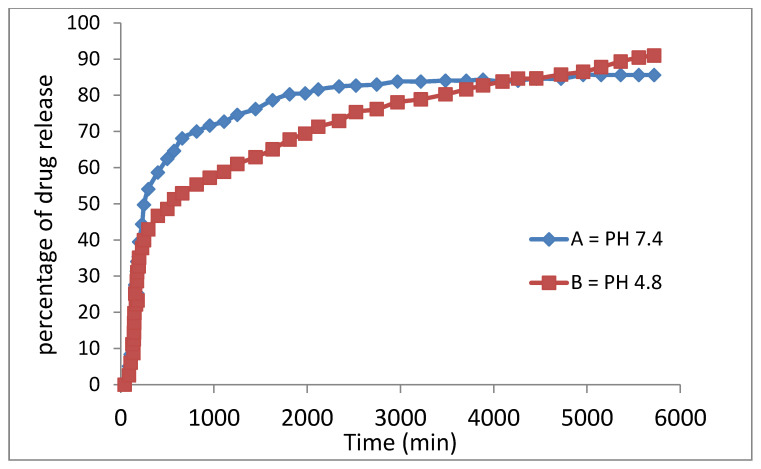
Release profiles of sorafenib from M-PEG–SO–ZLDH in phosphate-buffered solution at (**A**) pH 7.4 and (**B**) pH 4.8.

**Figure 12 polymers-12-02716-f012:**
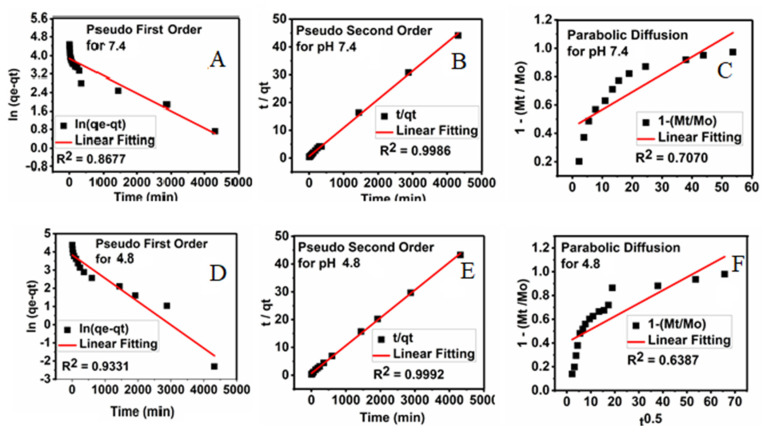
Fitting the data of sorafenib release from its M-PEG–SO–ZLDH nanoparticles into phosphate-buffered saline (PBS) solution to (**A**) the pseudo-first-order kinetic; (**B**) the pseudo-second-order kinetic; (**C**) the parabolic diffusion for pH 7.4; (**D**) the pseudo-first-order kinetic; (**E**) the pseudo-second-order kinetic; (**F**) the parabolic diffusion for pH 4.8. Abbreviations: t, time; q_t_, release at time t.

**Figure 13 polymers-12-02716-f013:**
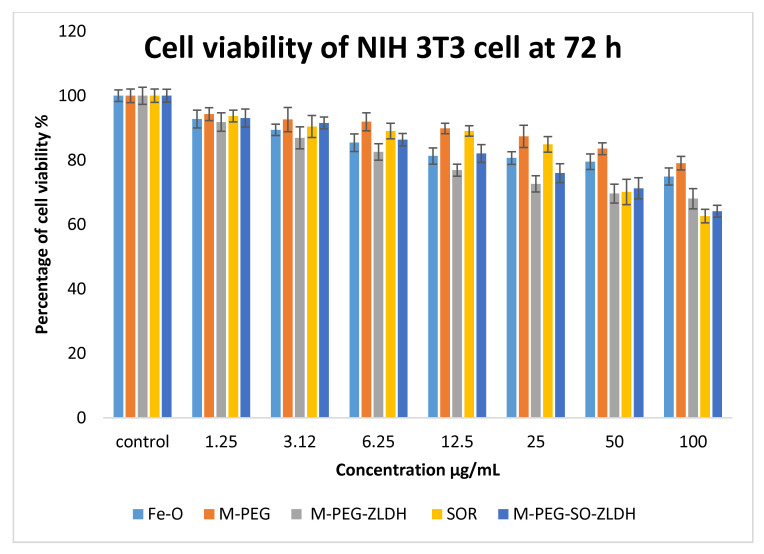
Cytotoxicity assay of Fe–O, M-PEG, M-PEG–ZLDH (the nanocarriers), pristine sorafenib, and M-PEG–SO–ZLDH against normal 3T3 cells at 72 h. Results were calculated as mean ± standard deviation for *n* = 3 independent experiments.

**Figure 14 polymers-12-02716-f014:**
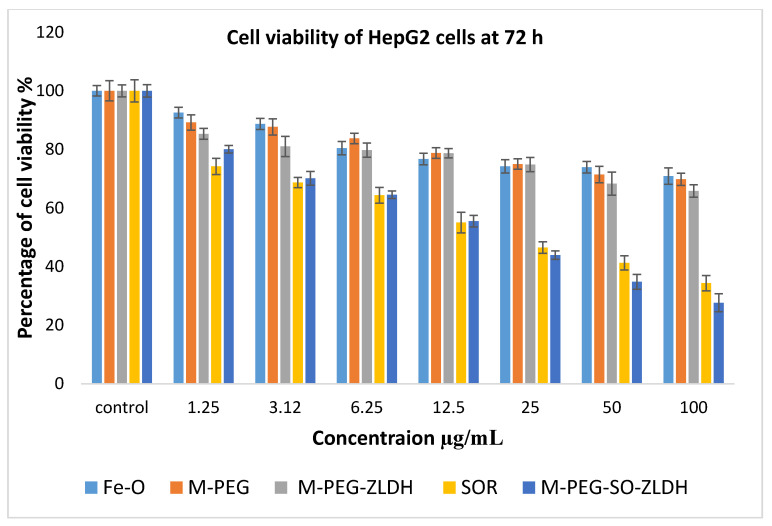
Cytotoxicity assay of Fe–O, M-PEG, M-PEG–ZLDH (the nanocarriers), pristine sorafenib, and M-PEG–SO–ZLDH (the drug delivery nanoparticles) against HepG2 cells at 72 h of incubation. Results were calculated as mean ± standard deviation for *n* = 3 independent experiments.

**Table 1 polymers-12-02716-t001:** The magnetic property of superparamagnetic iron oxide nanoparticles (SPIONs) and their M-PEG–SO–ZLDH.

Samples	M_s_ (emu/g)	M_r_ (emu/g)	H_ci_ (G)
SPIONs	73	2.77	13.65
M-PEG–SO–ZLDH	30	1.68	23.56

Notes: The data are presented in terms of emu/g·Oe^−1^ (mass magnetization. magnetic field). M_s_—saturation magnetization; M_r_—remanant magnetization; H_ci_—high coercivity.

**Table 2 polymers-12-02716-t002:** The correlation coefficient, rate constant, and half-life obtained by fitting the sorafenib release data into the PBS solution at pH 4.8 and pH 7.4.

Sample	Saturation Release	R^2^			Pseudo-Second- Order Rate Constant (k(mg/min))	t_1/2_
		Pseudo- First-Order	Pseudo-Second- Order	Parabolic Diffusion		
4.8	99.90	0.9331	0.9992	0.6387	4.32 × 10^−3^	134
7.4	97.96	0.8677	0.9986	0.7070	4.31 × 10^−3^	181

**Table 3 polymers-12-02716-t003:** The half-maximal inhibitory concentration (IC_50_) value for Fe–O, M-PEG, M-PEG–ZLDH (nanocarriers), pristine sorafenib, and M-PEG–SO–ZLDH samples tested on 3T3 and HepG2 cell lines.

Nanoparticles IC_50_ (μg/mL)	3T3 Fibroblast Cell	HepG2 Cells
Fe–O	N.C.	N.C.
M-PEG	N.C.	N.C.
M-PEG–ZLDH	N.C.	N.C.
SOR	N.C.	21.58
M-PEG–SO–ZLDH	N.C.	15.66

Abbreviations: N.C. = no cytotoxicity; IC_50_—half maximal inhibitory concentration.

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
