# Peer review of "Synthesis and Cytotoxicity Study of Magnetite Nanoparticles Coated with Polyethylene Glycol and Sorafenib–Zinc/Aluminium Layered Double Hydroxide"

_polymers, 2020, doi:10.3390/polym12112716_

Round 1
Reviewer 1 Report
Dear Authors
I have noticed a lot of presentation mistakes. Try to fix the manuscript according to the guideline of the journal.
1) Introduction is very extensive, I found many lines are not supporting the aim of the topic. Try to avoid the long stories in the original article.
2) Professional English editing required.
3) 3.10.1. Cytotoxicity studies on normal 3T3 fibroblast cells: try to revise this whole heading looks very difficult to understand.
Hope all correction will be great for this paper
Author Response
NO |
comment |
reply |
|
||
1 |
Introduction is very extensive, I found many lines are not supporting the aim of the topic. Try to avoid the long stories in the original article.
|
Thank you very much for your comments. Based on your suggestion, we have rewritten the Introduction and improved it. We have done the amendments and they are indicated by track changes. |
2 |
Professional English editing required. |
Thank you very much for your kind suggestion. The English language of the manuscript was revised and improved.
|
3 |
3.10.1. Cytotoxicity studies on normal 3T3 fibroblast cells: try to revise this whole heading looks very difficult to understand. |
Thank you very much for your kind comment. Based on your suggestion, the heading has been changed.
|

Reviewer 2 Report
The manuscript presents interesting findings in the investigation of the coated- and drug-loaded superparamagnetic iron oxide nanoparticles for cancer treatment. It is well structured and the scientific content is clear.
However, it is not clear what new findings are revealed in this manuscript, because the superparamagnetic iron oxide nanoparticles have been comprehensively studied by the authors in previously publications: https://doi.org/10.1016/j.aej.2020.09.061, https://doi.org/10.2147/IJN.S214923, 10.2147/IJN.S214923.
The abbreviations, such as ZLDH, SO, etc., should be clarified.
Conclusion should summarized what has been learned and why it is interesting and useful, but not describe the problem repeatedly. So, it should be carefully revised and rewritten.
Author Response
NO |
comment |
reply |
|
||
1 |
It is not clear what new findings are revealed in this manuscript, because the superparamagnetic iron oxide nanoparticles have been comprehensively studied by the authors in previously publications: https://doi.org/10.1016/j.aej.2020.09.061, https://doi.org/10.2147/IJN.S214923, 10.2147/IJN.S214923. |
Thank you very much for your comments. In our previous studies, the effects of other polymers (polyvinyl alcohol in https://doi.org/10.1016/j.aej.2020.09.061) and also other drugs effective in treating cancer (5-fluorouracil in https://doi.org/10.2147/IJN.S214923) were investigated. The current study is different from our previous studies mentioned by you. In the present study different polymer and anticancer drugs have been used and studied. In the present manuscript, we discussed the effects of polyethylene glycol polymer as the coating agent together with sorafenib as an anti-cancer drug co-coated with zinc/aluminum has been investigated. In fact, our goal in conducting these studies was to examine the effect of different polymers, drugs, and co-coatings factors to see if all the selected components could act as drug carriers and also to introduce different carriers for anticancer drugs that are suitable for targeted drug delivery systems.
|
2 |
The abbreviations, such as ZLDH, SO, etc., should be clarified. |
Thank you very much for your kind comment. We have done the corrections. The corrections are indicated in lines 23 and 24 (abstract).
|
3 |
Conclusion should summarized what has been learned and why it is interesting and useful, but not describe the problem repeatedly. So, it should be carefully revised and rewritten. |
Thank you very much for your kind comment. Based on your suggestion, we have rewritten the conclusion and improved it. The amendments are indicated by track changes. |

Round 2
Reviewer 1 Report
Dear Authors
It's improved very well. Please check minor English mistakes in the proof read carefully
Reviewer 2 Report
Thanks to the authors for making changes to the manuscript. The overall quality has improved and in my opinion the manuscript can be accepted for publication in Polymers.